# Temporal trend in the national and sub-national burden of cancers attributable to risk factors in Iran from 1990 to 2021: Findings from the global burden of disease study 2021

Seyede Maryam Mousavi[1,2◉], Sobhan Younesian[1,2◉], Saba Katebian[1], Ali Golestani[1], Shaghayegh Khanmohammadi[1,3], Sepehr Khosravi[1], Yasaman Etemadi[1], Nazila Rezaei[1], Sina Azadnajafabad [1]*, Bagher Larijani[4]*

1 Non-Communicable Diseases Research Center, Endocrinology and Metabolism Population Sciences Institute, Tehran University of Medical Sciences, Tehran, Iran, 2 School of Medicine, Tehran University of Medical Sciences, Tehran, Iran, 3 Research Center for Immunodeficiencies, Pediatrics Center of Excellence, Children's Medical Center, Tehran University of Medical Sciences, Tehran, Iran, 4 Endocrinology and Metabolism Research Center, Endocrinology and Metabolism Clinical Sciences Institute, Tehran University of Medical Sciences, Tehran, Iran

◉ These authors contributed equally to this work.
* sina.azad.u@gmail.com (SA); emrc@tums.ac.ir (BL)

## Abstract

### Background

Cancer is among leading causes of death globally and in Iran. However, studies exploring cancer risk factors trends in Iran are scarce. In this study, we provide estimations of risk-attributable cancer burden at the national and subnational levels in Iran from 1990 to 2021.

### Methods

This study utilized data from the Global Burden of Disease (GBD) 2021 Study to estimate cancer-related years of life lost (YLLs), years lived with disability (YLDs), disability-adjusted life years (DALYs), and deaths attributable to behavioral, metabolic, and environmental/occupational risks in Iran nationally and subnationally, from 1990 to 2021. Summary exposure values (SEV) were given to assess the level of exposure. All estimations were reported along with 95% uncertainty intervals (UI).

### Results

In 2021, 29.2% (95% UI: 22.9%–35.7%) of cancer deaths, equaling 16,893 (13,332–20,914) deaths and age-standardized rate of 22.66 (17.90–28.14), were attributable to risk factors in Iran. Since 1990, the number of risk-attributable cancer deaths increased by 192% (146% to 242%). Regarding attributable DALYs and deaths, the key risk factors were tobacco, dietary risks, and high body-mass index (BMI), with

**Data availability statement:** The data underlying the results presented in the study are publicly available from https://ghdx.healthdata.org/gbd-2021

**Funding:** The author(s) received no specific funding for this work.

**Competing interests:** The authors have declared that no competing interests exist.

high BMI and high fasting plasma glucose increasing by two-fold in DALYs. Tracheal, bronchus, and lung cancer, followed by colorectal cancer and stomach cancer, had the highest risk-attributable number of DALYs and deaths in both sexes. The risk-attributable age-standardized DALY rates for ovarian cancer [207% (87%–382%)], thyroid cancer [198% (74%–294%)], and multiple myeloma [192% (98%–349%)] showed the most significant increases.

## Conclusions

The all-age number of cancer deaths attributable to risk factors have increased in Iran. The age-standardized DALY rates attributable to high BMI and high FPG doubled from 1990 to 2021, indicating the emerging role of metabolic risk factors in cancer burden. These insights will guide effective cancer prevention strategies in Iran.

## 1. Introduction

Cancer remains a leading public health threat worldwide, causing 20 million new cases and 9.7 million deaths in 2022. Based on the Global Cancer Observatory (GCO) projections, over 35 million new cancer cases will occur in 2050 [1]. In 2019, 44% of cancer-related deaths were attributable to modifiable risk factors, with tobacco, alcohol, and high body mass index (BMI) as the leading risk factors globally [2]. In 2020, of the 5.28 million cancer-related deaths, 3.63 million deaths could have been averted worldwide through risk factor mitigation and early diagnosis [3].

Despite the higher incidence of cancer in high-income countries (HICs), cancer death rates are higher in low- and middle-income countries (LMICs). By 2030, up to three-fourths of all cancer deaths will happen in LMICs [4]. Iran, classified as an LMIC, had 137,138 incident cancer cases in 2022 [5,6], and this number is anticipated to increase by 110% until 2045 [7]. The Sustainable Development Goals (SDG), proposed by the United Nations in 2015, encompass reducing premature mortality caused by Non-Communicable Diseases (NCDs), including cancer, by one-third by 2030 as its Target 3.4 [8]. However, in many LMICs, national preventive cancer control plans do not exist. As a primary prevention strategy, reducing cancer risk factors is a major step towards this SDG target and should be undertaken at governmental, community, and individual levels [9]. Although many cancer risk factors are consistently reported across different countries, genetic, cultural, and socioeconomic variations between regions shape country-specific risk factors associated with various cancer types [10,11]. The World Health Organization (WHO) framework for national cancer control programs involves three planning steps, with the identification and prioritization of country-level risk factors as the first step [12]. Therefore, studies to estimate the relevance and significance of cancer-related risk factors in each country are mandated, as they aid in assessing progress toward SDG targets and inform planning for future policies.

Costs associated with cancer treatment are a substantial economic burden in Iran, underscoring the necessity of preventive strategies against cancer development [13].

The Iran National Cancer Control Program (IrNCCP), developed in 2013, is dedicated to cancer prevention, early detection, diagnosis, and treatment. Regarding cancer prevention, the initial goal is to determine priorities based on prevalent cancer types and the attributable risk factors at national and provincial levels [14]. A previous study indicated that in 2020, 33.8% of new cancer cases in Iran were attributable to preventable risk factors, with smoking, excess weight, and opium use identified as the leading risk factors [15]; in the same year, more than half of the years of life lost (YLLs) due to premature cancer deaths in Iran were avoidable through control measures, including elimination of the risk factors [3]. In particular, the prevalence of opium use among Iranian males is 10%, and it is projected that unless this risk factor is mitigated, one-third of the total incident cancer cases between 2020 and 2030 will be attributable to opium use [16]. Tobacco is yet another identified cancer risk factor in Iran, with 43% of cancer patients being tobacco users [17].

Currently available information on the relevant cancer risk factors in Iran is scarce and mainly comes from studies that focus on a single cancer type or risk factor [18]. To fill this gap of knowledge, we explored the burden of total and site-specific cancers attributable to risk factors and their patterns in Iran from 1990 to 2021 based on the Global Burden of Disease (GBD) 2021 study [19,20]. As the GBD 2021 study provided provincial information on cancer-related risk factors and their attributable burden in Iran, we also presented and compared subnational risk factors and their associated cancer types. By the means mentioned above, this study imparts updated information on cancer burden attributable to behavioral, metabolic, and environmental/occupational risk factors in Iran over 32 years. Furthermore, the role of sociodemographic differences in determining regional cancer risk factors was assessed by categorizing provinces based on their sociodemographic index (SDI) levels. The findings of this study will provide insights into the current status of risk-attributable cancer burden and introduce the related risk factors nationwide, guiding the forthcoming preventive strategies while evaluating the effectiveness of previously taken actions.

## 2. Materials and methods

### 2.1. Overview

The GBD study, conducted by the Institute for Health Metrics and Evaluation (IHME), provides insights into various health metrics, such as incidence, prevalence, mortality, YLLs, years lived with disability (YLDs), and disability-adjusted life years (DALYs), across 204 countries and 811 sub-national locations from 1990 to 2021. GBD 2021, containing two new causes of death and one non-fatal cause since the last iteration, is the most up-to-date version of GBD [21]. The 2021 GBD study offers annual estimates on the burden of 371 diseases and injuries [21], 288 causes of death [20], and 88 risk factors [19]. The cancer estimates in GBD 2021 differ from GBD 2019 for several reasons. First, the new data and updates from registries included a significant amount of new pediatric cancer registry data, which informed the MIRs. Second, GBD 2021 introduced new cancers, requiring a shift from "other malignant neoplasms" to these new causes. These additions necessitated the development of new models and, in some instances, new estimation approaches due to the rarity and limited data of these cancers. The GBD study and current report adhere to the Guidelines for Accurate and Transparent Health Estimates Reporting (GATHER) throughout its analytical process [22]. All data used in this study are publicly accessible via the GBD Compare (https://vizhub.healthdata.org/gbd-compare) and GBD Results (http://ghdx.healthdata.org/gbd-results-tool) web pages.

### 2.2. Definitions

The GBD study categorizes diseases and injuries hierarchically, from level 1 to level 4, and further into levels 5 and 6 for sequelae. NCDs, a level 1 cause, comprise 12 categories. Cancer, also termed neoplasms, is a level 2 cause and includes 34 level 3 cancers. Cervical, uterine, and ovarian cancers are restricted to females, while prostate and testicular cancers are limited to males. Details on age restrictions, modeling strategies, covariates for each cancer type, and International Classification of Diseases (ICD) codes have been described elsewhere [19,21].

The GBD project employs comparative risk assessment (CRA) to estimate the contribution of risk factors to disease burden. Attributable burden, the main risk assessment used by GBD, targets the change in current disease burden if past populations had adopted a counterfactual risk exposure. This quantification involves various scenarios, including the theoretical minimum risk exposure level (TMREL), representing the lowest achievable risk exposure. TMREL is used to calculate the population attributable fraction (PAF), which measures the proportional reduction in a disease's burden within a specific year if a counterfactual exposure to a risk factor had occurred [20].

### 2.3. Data source

In GBD 2021, data on non-fatal causes were obtained from systematic reviews of published literature, survey data, disease registers, and hospital data. The GBD cause of death (CoD) database contains cancer mortality data from various sources like vital registration, verbal autopsy, and cancer registries. The GBD 2021 utilized mortality-to-incidence ratios (MIRs) to transform cancer incidence data to mortality estimates. MIRs were estimated using cancer registry data matched by cancer type, age, sex, year, and location. A spatiotemporal Gaussian process regression (ST-GPR) model incorporated covariates such as age, sex, and the Healthcare Access and Quality Index, with smoothing across time, age, and geography. Additionally, adjustments were made for rare cancers and outliers to ensure reliable estimates across all cancer types and demographic groups. Detailed explanation on the estimation of MIRs are provided elsewhere [19]. Before entering the CoD database, cancer registry data undergo multiple processing steps, such as mapping to GBD causes, age/sex splitting, and cause disaggregation, as detailed elsewhere [19,21]. In GBD 2021, from a total of 544 input citations from Iran, 34 citations have been used to estimate cancer CoD, and an additional 20 citations were input data for the estimation of non-fatal health outcomes of cancer. These input sources, along with their metadata, can be explored from the GBD 2021 Sources Tool (https://ghdx.healthdata.org/gbd-2021/sources). Furthermore, all data sources are accessible via the Global Health Data Exchange (GHDx) web tool (http://ghdx.healthdata.org). Also, the code used to perform all estimations are publicly available at https://github.com/ihmeuw/ihme-modeling/tree/main/gbd_2021. The data for the purpose of this research work was accessed on October 20, 2024.

### 2.4. Statistical analysis

The PAF for each risk-outcome was multiplied by the burden measures to calculate the risk factor attributable burden, including deaths, YLDs, YLLs, and DALYs. YLDs were determined by multiplying the prevalence of general and procedure sequelae of each cancer type by their respective disability weights before summing them. YLLs were calculated by multiplying the life expectancy of each age group by the number of deaths in that age group. DALYs were then obtained as the sum of YLDs and YLLs [19–21]. The SDI for a location was calculated as the geometric mean of income per capita, average years of education for individuals aged 15 and older, and the total fertility rate for females under 25, scaled from 0 to 1 [23,24]. In 2021, the SDI of various provinces in Iran ranged from 0.55 to 0.78. These provinces were subsequently categorized into five quintiles. The Summary Exposure Value (SEV) measures exposure to a risk factor by considering its risk level and the severity of its outcomes [20]. The 95% uncertainty interval (UI) for each metric was calculated using the 2.5th and 97.5th percentiles from the uncertainty distribution, based on 1000 draws from the posterior distribution [19–21].

To explore the relationship between SEV values and age-standardized DALY rates attributable to the leading risk factors, the annualized rate of change (ARC) was estimated from 1990 to 2021 as below:

$$\sqrt[n]{\frac{value_2}{value_1}} - 1$$

Where $value_1$ and $value_2$ are the estimated values at the beginning and end of the time interval, n equals the length of the time interval. ARC of SEV a specific risk factor was compared to the ARC of age-standardized DALY rate attributable to the same risk factor over the same time period.

To discover the non-linear relationship between SDI and percentage of cancer DALYs attributable to risk factors, Estimated Scatterplot Smoothing (LOESS) regression was performed, with SDI values of provinces throughout the study period as the independent variable and the age-standardized percentage of cancer DALYs attributed to all and the top-five risk factors as the dependent variables.

Data analysis and visualization were performed with R (version 4.4.0) using *ggplot2* package Tableau Desktop (version 2019.4), and Python programming language (version 3.12.4) using *pandas* library.

### 2.5. Ethics statement

This study is a secondary analysis of publicly available, previously collected, and de-identified data from the Global Burden of Disease Study 2021. Therefore, participant consent is not applicable. This study received approval from the institutional review board of the Endocrinology and Metabolism Research Institute at Tehran University of Medical Sciences (IR. TUMS.EMRI.REC.1401.165). The findings are derived from estimates provided by the GBD 2021 study and comply with applicable guidelines and regulations. The data that support the findings of this study are available from the IHME through https://vizhub.healthdata.org/gbd-compare and http://ghdx.healthdata.org/gbd-results-tool.

## 3. Results

### 3.1. Overview

In 2021, a total number of 16,893 (95% UI: 13,332–20,914) cancer deaths attributable to risk factors, which was 29.2% (22.9–35.7) of all cancer deaths, occurred in Iran. The total number of risk-attributable cancer deaths has increased by 192% (146–242). In females, the number of risk factor-attributable cancer deaths has risen from 1,758 (1,207–2,376) in 1990–5,933 (3,863–7,802) in 2021; in males, the total number of risk factor-attributable deaths was 4,019 (3,241–5,312) in 1990 and increased to 10,960 (9,195–13,183) in 2021 (Table 1). In 2021, the cancer age-standardized death rate (ASDR) attributable to risk factors was 22.7 (17.9–28.1) for both sexes, and the change in ASDR from 1990 to 2021 was not significant in either sex. Overall, the age-standardized DALY rate declined from 1990 to 2003, but an inclining trend took over from 2004 to 2019; in 2020, there was a relatively sharp decrease in the age-standardized DALY rate (S1 Fig). A similar trend was observed for the death rates. In 2021, the total number of risk factor-attributable cancer DALYs was 447,269 (350,569–550,594) for both sexes, which equals 27.2% (21.3–33.6) of total DALYs associated with cancers. The age-standardized DALY rate was 544.42 (427.39–669.64) per 100,000 in both sexes in 2021 and had not significantly changed compared to 1990 (Table 1); moreover, the percentage of cancer DALYs attributable to risk factors had not significantly increased since 1990 (S2 Fig). Overall, risk-attributable death and DALY rates were higher in the older age groups than the younger. Similar to 1990, the highest risk-attributable cancer death rate was in the > 80-year-old age group [275.1 (204.8–352.2) per 100,000] in 2021. In 1990 and 2021, the highest DALY rate was seen in the 70–74 and 75–79 age groups, respectively (Fig 1).

### 3.2. Risk factors at the national level

At the national level, behavioral risks were the leading level 1 risk factor for cancer deaths and DALYs in 2021, followed by metabolic and environmental/occupational risks in females, males, and both sexes (Table 1). In total, eleven level 2 risk factors were identified, and the trends of their attributable cancer burden are depicted in Figure 2. Among level 2 risk factors, tobacco, dietary risks, and high BMI were the top three risk factors for cancer deaths and DALYs in both sexes, followed by high FPG and air pollution. In females, dietary risks and high BMI were the top contributors to cancer deaths and DALYs; however, high FPG and tobacco were the third leading risk factors of all-age cancer death and DALY numbers in females, respectively. In males, tobacco and dietary risks had the highest attributable all-age cancer DALY and death numbers, and the third leading risk factor was air pollution for deaths and high BMI for DALYs (S1 Table). The contribution

**Table 1. DALYs, deaths, YLDs, and YLLs of cancer attributable to all risk factors and level 1 risk factors among females, males, and both sexes in Iran in 1990 and 2021 and their percent change.**

| Risk factors | Measure | Age, Metric | Year 1990 Both | Year 1990 Female | Year 1990 Male | 2021 Both | 2021 Female | 2021 Male | Percent Change (1990-2021) Both | Percent Change (1990-2021) Female | Percent Change (1990-2021) Male |
|---|---|---|---|---|---|---|---|---|---|---|---|
| All risk factors | DALYs | All age number | 169057.47 (135238.74 to 218941.85) | 55645.71 (38161.28 to 74699.43) | 113411.76 (91220.05 to 149142.32) | 447268.61 (350568.82 to 550593.51) | 165709.97 (108823.27 to 218899.75) | 281558.64 (236594.8 to 340049.01) | 164.57% (124.58% to 207.05%) | 197.79% (138.06% to 259.31%) | 148.26% (103.2% to 194.34%) |
| | | Age-standardized rate (per 100,000) | 584.85 (467.81 to 762.74) | 387.69 (264.57 to 524.82) | 766.6 (619.83 to 1010.0) | 544.42 (427.39 to 669.64) | 395.53 (258.69 to 523.92) | 696.22 (585.81 to 842.78) | -6.91% (-21.36% to 8.15%) | 2.02% (-18.42% to 23.12%) | -9.18% (-25.19% to 7.75%) |
| | Deaths | All age number | 5778.17 (4639.16 to 7583.0) | 1758.85 (1207.42 to 2375.57) | 4019.31 (3240.63 to 5312.21) | 16893.25 (13331.94 to 20914.49) | 5933.16 (3863.45 to 7801.84) | 10960.09 (9195.23 to 13182.79) | 192.36% (146.15% to 241.64%) | 237.33% (168.59% to 306.94%) | 172.69% (125.22% to 224.12%) |
| | | Age-standardized rate (per 100,000) | 23.34 (18.59 to 30.75) | 14.62 (9.93 to 19.96) | 31.81 (25.62 to 42.21) | 22.66 (17.9 to 28.14) | 15.76 (10.22 to 20.8) | 29.66 (24.84 to 35.87) | -2.94% (-18.34% to 12.71%) | 7.81% (-14.25% to 29.48%) | -6.77% (-23.14% to 10.58%) |
| | YLDs | All age number | 2845.35 (1847.75 to 4070.69) | 1311.37 (681.54 to 2036.5) | 1533.98 (1069.24 to 2137.58) | 13559.52 (8146.46 to 20113.56) | 7782.84 (3716.82 to 12517.0) | 5776.69 (4051.35 to 7933.35) | 376.55% (286.23% to 462.01%) | 493.49% (359.83% to 619.06%) | 276.58% (214.66% to 346.92%) |
| | | Age-standardized rate (per 100,000) | 10.05 (6.61 to 14.41) | 9.16 (4.71 to 14.01) | 10.9 (7.64 to 15.19) | 16.3 (10.0 to 24.07) | 18.04 (8.56 to 28.91) | 14.62 (10.24 to 20.11) | 62.08% (30.17% to 90.13%) | 96.99% (53.01% to 139.18%) | 34.13% (11.69% to 58.62%) |
| | YLLs | All age number | 166212.12 (133170.12 to 215972.6) | 54334.34 (37506.63 to 72761.98) | 111877.78 (90113.1 to 147301.29) | 433709.08 (341463.17 to 535885.13) | 157927.13 (104944.03 to 207852.0) | 275781.95 (232326.61 to 333612.37) | 160.94% (121.64% to 203.05%) | 190.66% (132.45% to 251.2%) | 146.5% (101.7% to 192.79%) |
| | | Age-standardized rate (per 100,000) | 574.8 (460.34 to 751.92) | 378.53 (259.51 to 511.17) | 755.69 (610.9 to 997.17) | 528.13 (416.41 to 652.31) | 377.49 (248.29 to 497.18) | 681.59 (573.3 to 826.37) | -8.12% (-22.28% to 6.74%) | -0.27% (-20.19% to 20.22%) | -9.81% (-25.76% to 7.1%) |
| Behavioral risks | DALYs | All age number | 142892.05 (110809.4 to 192368.52) | 42275.63 (26733.56 to 60877.75) | 100616.42 (79386.99 to 135717.35) | 326480.52 (255929.1 to 414441.41) | 98827.69 (59362.83 to 139891.98) | 227652.83 (190099.36 to 280800.23) | 128.48% (98.77% to 161.75%) | 133.77% (92.47% to 181.73%) | 126.26% (91.59% to 167.67%) |
| | | Age-standardized rate (per 100,000) | 494.79 (385.25 to 668.34) | 291.66 (181.89 to 420.96) | 681.73 (538.2 to 918.24) | 395.46 (310.18 to 501.52) | 230.17 (139.87 to 323.29) | 563.8 (469.11 to 695.23) | -20.07% (-30.47% to -8.48%) | -21.08% (-33.71% to -5.26%) | -17.3% (-29.99% to -1.79%) |
| | Deaths | All age number | 4898.23 (3817.91 to 6661.06) | 1312.31 (819.91 to 1899.37) | 3585.92 (2816.76 to 4858.93) | 12273.34 (9683.94 to 15558.44) | 3380.83 (2108.91 to 4728.31) | 8892.51 (7379.37 to 11065.44) | 150.57% (118.62% to 187.51%) | 157.62% (117.62% to 207.65%) | 147.98% (109.5% to 194.86%) |
| | | Age-standardized rate (per 100,000) | 19.67 (15.21 to 26.94) | 10.78 (6.71 to 15.7) | 28.28 (22.21 to 38.36) | 16.38 (12.89 to 20.86) | 8.82 (5.5 to 12.42) | 24.05 (19.92 to 30.04) | -16.72% (-27.47% to -4.4%) | -18.21% (-30.19% to -3.45%) | -14.97% (-28.17% to 0.41%) |

*(Continued)*

| Risk factors | Measure | Age, Metric | Year 1990 Both | Female | Male | 2021 Both | Female | Male | Percent Change (1990-2021) Both | Female | Male |
|---|---|---|---|---|---|---|---|---|---|---|---|
| | YLDs | All age number | 2382.43 (1517.96 to 3466.0) | 1006.78 (473.65 to 1660.02) | 1375.65 (956.97 to 1940.98) | 9525.38 (5343.87 to 14616.85) | 4879.91 (1814.66 to 8427.02) | 4645.47 (3186.4 to 6491.01) | 299.82% (226.64% to 370.61%) | 384.7% (241.68% to 494.21%) | 237.69% (187.75% to 297.25%) |
| | | Age-standardized rate (per 100,000) | 8.37 (5.41 to 12.12) | 6.87 (3.22 to 11.15) | 9.77 (6.79 to 13.67) | 11.29 (6.51 to 17.05) | 10.89 (4.18 to 18.57) | 11.79 (8.2 to 16.43) | 34.88% (11.47% to 57.66%) | 58.59% (13.57% to 93.05%) | 20.77% (3.32% to 41.07%) |
| | YLLs | All age number | 140509.62 (109245.59 to 189714.04) | 41268.84 (26257.14 to 59388.66) | 99240.78 (78076.1 to 133942.6) | 316955.14 (250119.81 to 400182.47) | 93947.79 (57475.55 to 132101.72) | 223007.36 (186641.46 to 275955.62) | 125.58% (96.38% to 157.94%) | 127.65% (90.38% to 173.04%) | 124.71% (90.42% to 166.0%) |
| | | Age-standardized rate (per 100,000) | 486.42 (379.06 to 658.98) | 284.79 (178.67 to 409.78) | 671.96 (530.54 to 905.21) | 384.17 (302.99 to 485.33) | 219.28 (135.89 to 306.4) | 552.01 (460.65 to 682.86) | -21.02% (-31.21% to -9.8%) | -23.0% (-35.15% to -7.68%) | -17.85% (-30.54% to -2.42%) |
| Environmental/Occupational risks | DALYs | All age number | 24208.82 (16421.48 to 33040.01) | 4874.97 (2980.65 to 7311.18) | 19333.86 (13106.69 to 26737.92) | 63656.78 (44318.05 to 83178.45) | 17379.15 (10885.28 to 23811.26) | 46277.63 (33365.53 to 59918.59) | 162.95% (106.28% to 243.48%) | 256.5% (145.21% to 418.66%) | 139.36% (79.74% to 214.18%) |
| | | Age-standardized rate (per 100,000) | 83.17 (56.18 to 114.36) | 35.34 (21.53 to 52.63) | 127.18 (85.53 to 176.35) | 77.1 (53.2 to 101.44) | 42.37 (26.4 to 57.97) | 112.18 (79.99 to 145.71) | -7.3% (-28.01% to 21.64%) | 19.91% (-16.99% to 74.45%) | -11.79% (-33.76% to 16.37%) |
| | Deaths | All age number | 834.58 (561.55 to 1152.27) | 165.19 (100.18 to 246.46) | 669.39 (448.36 to 934.7) | 2412.34 (1641.14 to 3200.11) | 675.86 (415.99 to 927.14) | 1736.48 (1222.81 to 2274.85) | 189.05% (123.95% to 280.44%) | 309.15% (183.55% to 497.42%) | 159.41% (94.04% to 242.9%) |
| | | Age-standardized rate (per 100,000) | 3.33 (2.23 to 4.64) | 1.45 (0.87 to 2.16) | 5.14 (3.41 to 7.18) | 3.21 (2.15 to 4.28) | 1.84 (1.12 to 2.54) | 4.6 (3.2 to 6.05) | -3.72% (-25.59% to 27.09%) | 26.76% (-11.32% to 85.99%) | -10.5% (-32.59% to 19.1%) |
| | YLDs | All age number | 203.36 (128.07 to 304.9) | 39.87 (21.62 to 62.44) | 163.5 (104.79 to 251.45) | 583.02 (362.28 to 836.6) | 154.21 (86.62 to 232.46) | 428.82 (270.49 to 605.75) | 186.69% (125.89% to 272.11%) | 286.79% (166.23% to 458.06%) | 162.28% (99.4% to 246.04%) |
| | | Age-standardized rate (per 100,000) | 0.74 (0.47 to 1.13) | 0.31 (0.17 to 0.5) | 1.14 (0.72 to 1.75) | 0.73 (0.45 to 1.05) | 0.39 (0.22 to 0.59) | 1.07 (0.67 to 1.53) | -1.39% (-22.47% to 29.67%) | 24.94% (-13.98% to 81.08%) | -5.82% (-29.15% to 24.09%) |
| | YLLs | All age number | 24005.46 (16287.02 to 32742.93) | 4835.1 (2951.37 to 7255.22) | 19170.36 (13015.86 to 26531.46) | 63073.75 (43875.8 to 82488.85) | 17224.94 (10792.43 to 23575.04) | 45848.81 (33061.35 to 59380.95) | 162.75% (106.18% to 243.34%) | 256.25% (145.03% to 418.26%) | 139.17% (79.55% to 214.02%) |
| | | Age-standardized rate (per 100,000) | 82.43 (55.69 to 113.27) | 35.02 (21.3 to 52.19) | 126.03 (84.78 to 174.92) | 76.37 (52.69 to 100.54) | 41.98 (26.14 to 57.36) | 111.1 (79.28 to 144.36) | -7.35% (-28.04% to 21.62%) | 19.86% (-17.04% to 74.39%) | -11.85% (-33.81% to 16.28%) |

(Continued)

Table 1. (Continued)

| Risk factors | Measure | Age, Metric | Year 1990 Both | Female | Male | 2021 Both | Female | Male | Percent Change (1990-2021) Both | Female | Male |
|---|---|---|---|---|---|---|---|---|---|---|---|
| Metabolic risks | DALYs | All age number | 20470.22 (7477.21 to 32806.83) | 11491.2 (4504.37 to 18951.31) | 8979.02 (3381.1 to 14551.34) | 116123.75 (37867.2 to 193645.66) | 63512.63 (19837.2 to 105683.25) | 52611.12 (17358.8 to 89768.72) | 467.28% (332.39% to 559.86%) | 452.71% (293.18% to 558.65%) | 485.93% (364.91% to 607.99%) |
| | | Age-standardized rate (per 100,000) | 71.16 (24.23 to 116.63) | 83.11 (31.54 to 139.01) | 60.31 (20.75 to 99.16) | 144.22 (45.55 to 243.49) | 157.79 (48.26 to 264.72) | 130.62 (41.69 to 224.05) | 102.67% (61.59% to 133.65%) | 89.85% (43.16% to 125.52%) | 116.57% (76.64% to 156.52%) |
| | Deaths | All age number | 697.03 (233.08 to 1146.88) | 386.83 (144.36 to 648.31) | 310.2 (102.91 to 513.67) | 4474.22 (1367.62 to 7628.18) | 2422.59 (731.54 to 4030.2) | 2051.63 (627.52 to 3558.83) | 541.9% (404.44% to 637.16%) | 526.27% (373.48% to 639.85%) | 561.39% (434.72% to 687.69%) |
| | | Age-standardized rate (per 100,000) | 2.9 (0.91 to 4.86) | 3.3 (1.16 to 5.56) | 2.51 (0.75 to 4.2) | 6.09 (1.81 to 10.44) | 6.59 (1.95 to 11.0) | 5.6 (1.66 to 9.78) | 110.27% (71.7% to 138.89%) | 99.46% (56.29% to 134.84%) | 122.67% (83.25% to 161.84%) |
| | YLDs | All age number | 481.68 (159.28 to 847.89) | 328.44 (102.53 to 592.1) | 153.24 (56.23 to 268.68) | 4680.81 (1342.7 to 8322.0) | 3314.99 (855.63 to 5954.92) | 1365.82 (490.47 to 2391.1) | 871.77% (667.11% to 1039.38%) | 909.31% (652.33% to 1145.03%) | 791.29% (635.64% to 984.15%) |
| | | Age-standardized rate (per 100,000) | 1.75 (0.54 to 3.11) | 2.48 (0.73 to 4.47) | 1.11 (0.38 to 1.95) | 5.83 (1.66 to 10.42) | 8.18 (1.97 to 14.89) | 3.45 (1.2 to 6.08) | 232.54% (169.08% to 287.66%) | 230.43% (156.19% to 305.48%) | 212.07% (158.9% to 274.88%) |
| | YLLs | All age number | 19988.54 (7347.68 to 32059.43) | 11162.76 (4392.84 to 18298.95) | 8825.78 (3317.07 to 14298.02) | 111442.94 (36456.1 to 186744.73) | 60197.64 (18976.46 to 98904.24) | 51245.3 (16886.85 to 87240.67) | 457.53% (327.62% to 545.99%) | 439.27% (285.09% to 540.65%) | 480.63% (360.68% to 601.34%) |
| | | Age-standardized rate (per 100,000) | 69.41 (23.75 to 113.73) | 80.64 (30.74 to 134.37) | 59.21 (20.33 to 97.16) | 138.39 (43.88 to 233.46) | 149.61 (46.22 to 247.2) | 127.17 (40.58 to 217.63) | 99.39% (60.18% to 129.04%) | 85.53% (39.86% to 119.95%) | 114.79% (75.17% to 154.39%) |

DALYs: Disability-Adjusted Life Years. YLDs: Years Lived with Disability. YLLs: Years of Life Lost.

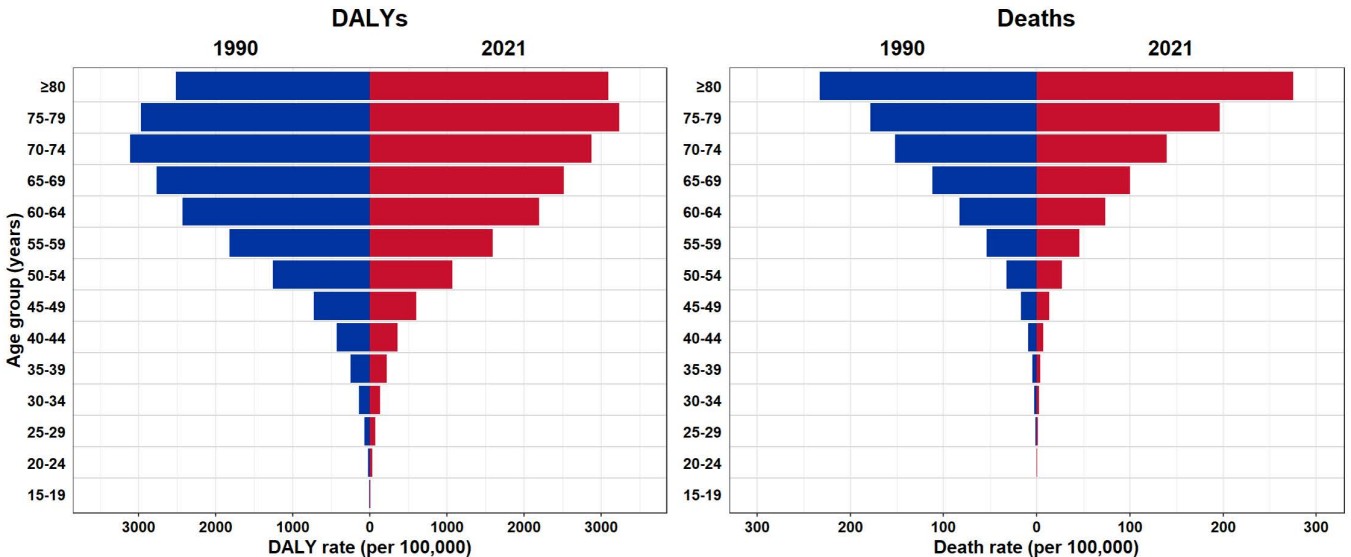

**Fig 1. DALY and death rates of cancer attributable to risk factors by different age groups in Iran in 1990 and 2021.**

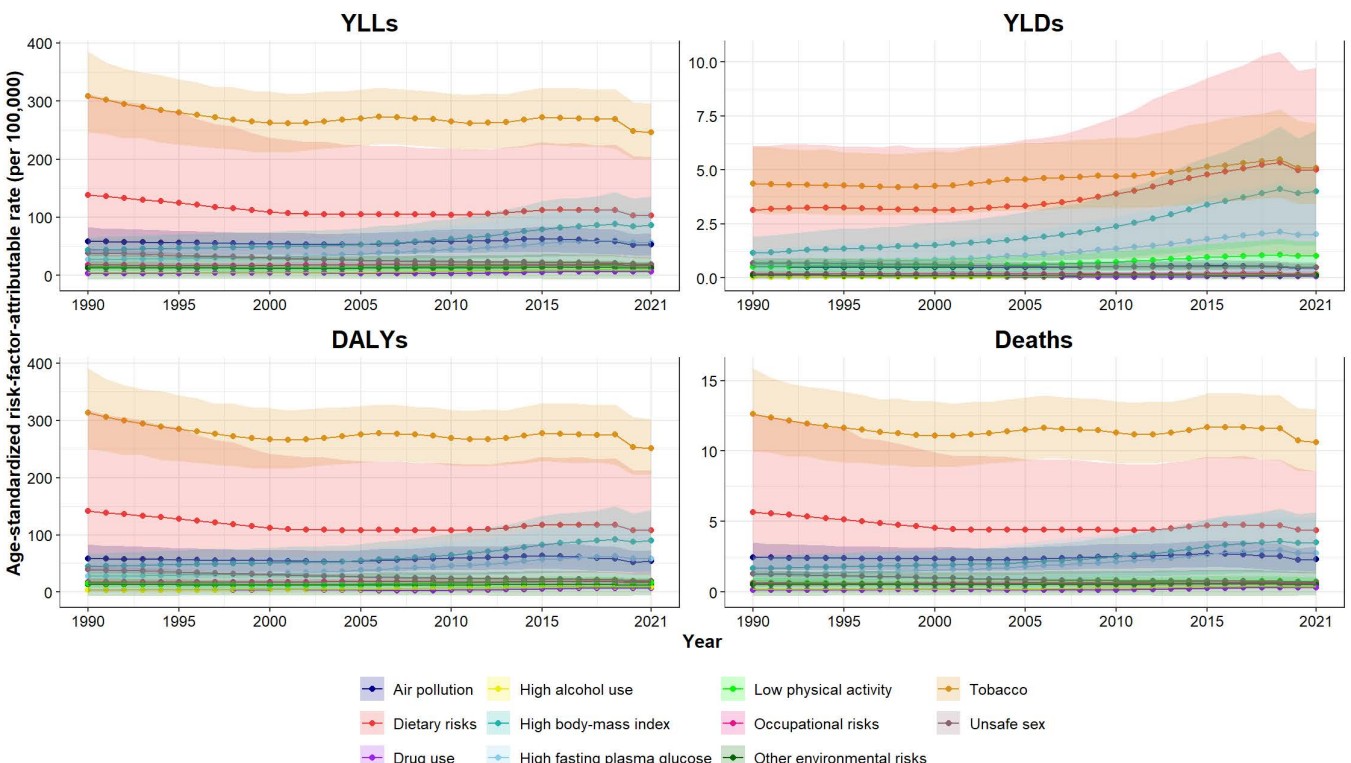

**Fig 2. Trends of age-standardized DALY, death, YLD, and YLL rates of cancer attributable to level 2 risk factors in Iran from 1990 to 2021.**

of various risk factors to cancer DALYs was different between males and females throughout the 32-year study period (S3 Figure). In 2021, the age-standardized death and DALY rates attributable to the top five cancer risk factors for both sexes had declined since 1990, except for high BMI and high FPG, which had increased. The high BMI and high FPG-attributable age-standardized DALY rates in both sexes elevated by 101% (55–137) and 113% (78–149), respectively from 1990 to 2021. A similar pattern was observed among behavioral risks in males, such as drug use and alcohol use, according to the attributable DALYs (S1 Figure).

### 3.3. Risk factors at the subnational level

In 2021, the highest age-standardized DALY rates attributable to all cancer risk factors were 705.93 (539.15–929.93) in Ardebil and 678.65 (519.31–814.91) in Golestan and the lowest were 358.20 (274.55–453.20) in Hormozgan and 395.00 (299.91–540.58) in Kohgiluyeh and Boyer-Ahmad. West-Azarbayejan had the highest risk factor-attributable cancer deaths, which was 29.43 (22.59–37.79) in 2021, followed by 29.25 (22.48–38.46) in Ardebil. Similar to DALYs, the lowest attributable death rates were seen in Hormozgan and Kohgiluyeh and Boyer-Ahmad. Cancer-related burden attributable to level 1 risk factors at the subnational level are provided in S2 Table. Overall, behavioral risks were the top risk factor of cancer DALYs in all provinces from 1990 to 2021. In females, a higher proportion of risk factor-attributable cancer DALYs were attributed to metabolic risks, compared to males in 2021 in all provinces; however, the contribution of metabolic risk factors to cancer DALYs has increased in both females and males from 1990 to 2021 (S4 Figure).

The proportion cancer burden attributable to each level 2 risk factor at the subnational level in 2021 is shown in S5 Figure. In general, tobacco, followed by dietary risks and high BMI, are the leading risk factors across all provinces. In 2021, the highest age-standardized cancer DALY rates attributable to tobacco, dietary risks, and air pollution were observed in Ardebil, East Azarbayejan, and Bushehr, respectively, while Tehran had the highest rates for high BMI and high FPG (Figure 3). Among these top-five level 2 risk factors, cancer death rates attributable to high BMI and high FPG have surged the most markedly from 1990 to 2021 in both sexes in all provinces. Ilam, a high-middle SDI province, has experienced the greatest rise in both high BMI- and high FPG-attributable cancer death rates between 1990 and 2021 (Figure 4). Among the top four risk factors of cancer-related DALYs, the burden of high BMI and high FPG has increased across provinces. Still, cancer DALYs attributable to tobacco and dietary risks have declined from 1990 to 2021.

### 3.4. Cancer types

The leading cancer types for risk factor-attributable deaths in 2021 were tracheal, bronchus, and lung cancer, with an all-age death count of 5,194 (4,497–5,897) in both sexes, followed by colon and rectum cancer and stomach cancer. In the same order, these three cancer types also led in all-age DALY numbers for males. For females, the top cancer types in terms of all-age DALY numbers were colon and rectum cancer, breast cancer, and tracheal, bronchus, and lung cancer. The risk-attributable age-standardized DALY rates related to ovarian cancer [207% (87–382)], thyroid cancer [198% (74–294)], and multiple myeloma [192% (98–349)] had the greatest surges from 1990 to 2021 (S3 Table). After tracheal, bronchus, and lung cancer, the second leading cancers attributable to tobacco were breast cancer in females and larynx cancer in males, based on both age-standardized death and DALY rates. In females and males, colon and rectum cancer was responsible for the greatest proportion of age-standardized DALY rate attributable to dietary risks, followed by breast cancer in females and stomach cancer in males. The highest age-standardized DALY rate attributable to high BMI in females was related to colon and rectum cancer and breast cancer (Figure 5 and S2 Table).

### 3.5. SEV

Figure 6 demonstrates the variations in DALY rates attributable to the top four cancer risk factors in relation to changes in their SEV. From 1990 to 2021, the SEV of high BMI and high FPG increased across all provinces. Conversely, tobacco and dietary risks predominantly exhibited declining of SEV. Overall, the ARC of SEV was positively related to ARC of

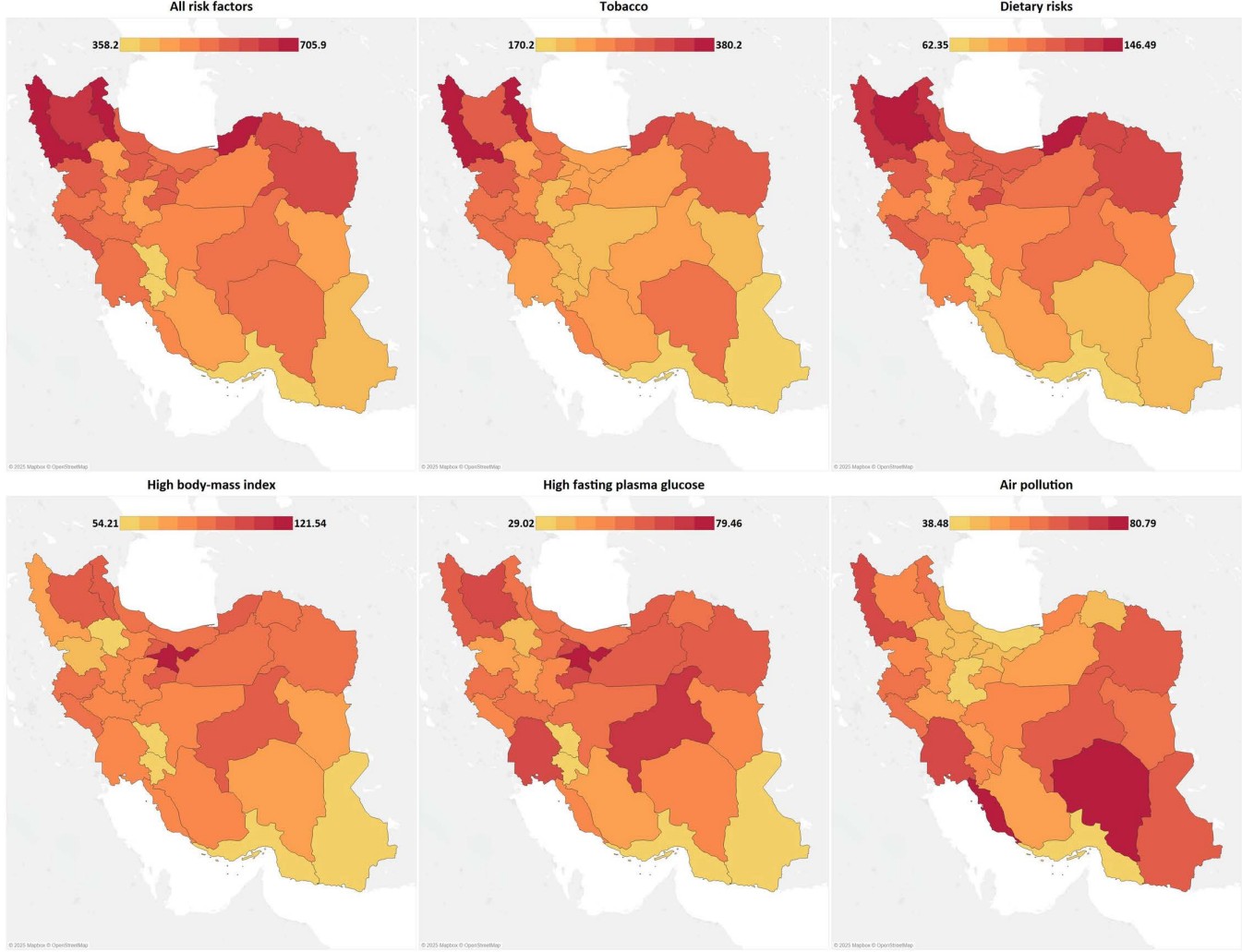

**Fig 3. Age-standardized DALY rates for all cancers attributable to all and the top five risk factors at the subnational level in 2021.**

cancer DALYs for tobacco, dietary risks, high BMI, and high FPG. As evident in Figure 6, the reduction in tobacco SEV has been confined to the extremes of SDI values in low and high SDI regions. Additionally, the pattern of change in dietary risks SEV in high SDI provinces mirrors the national trend.

### 3.6. The relationship between SDI and PAF for risk factors

Figure 7 illustrates the results of the LOESS analysis. As indicated, the age-standardized percentage of DALYs attributed to all risk factors increased with higher SDI values. Although tobacco showed an overall negative relationship between SDI and the age-standardized attributable percentage of DALYs, also known as PAF, this relationship reversed within the SDI range of about 0.5 to 0.6. Regarding dietary risks, there is a negative relationship between the PAF and SDI within the early SDI range, followed by a transition to a positive relationship in the higher SDI values. High BMI and high FPG exhibit an upward trend in the PAF with increasing SDI values. Moving to air pollution, a negative relationship between PAF and SDI becomes evident in the upper range of SDI values.

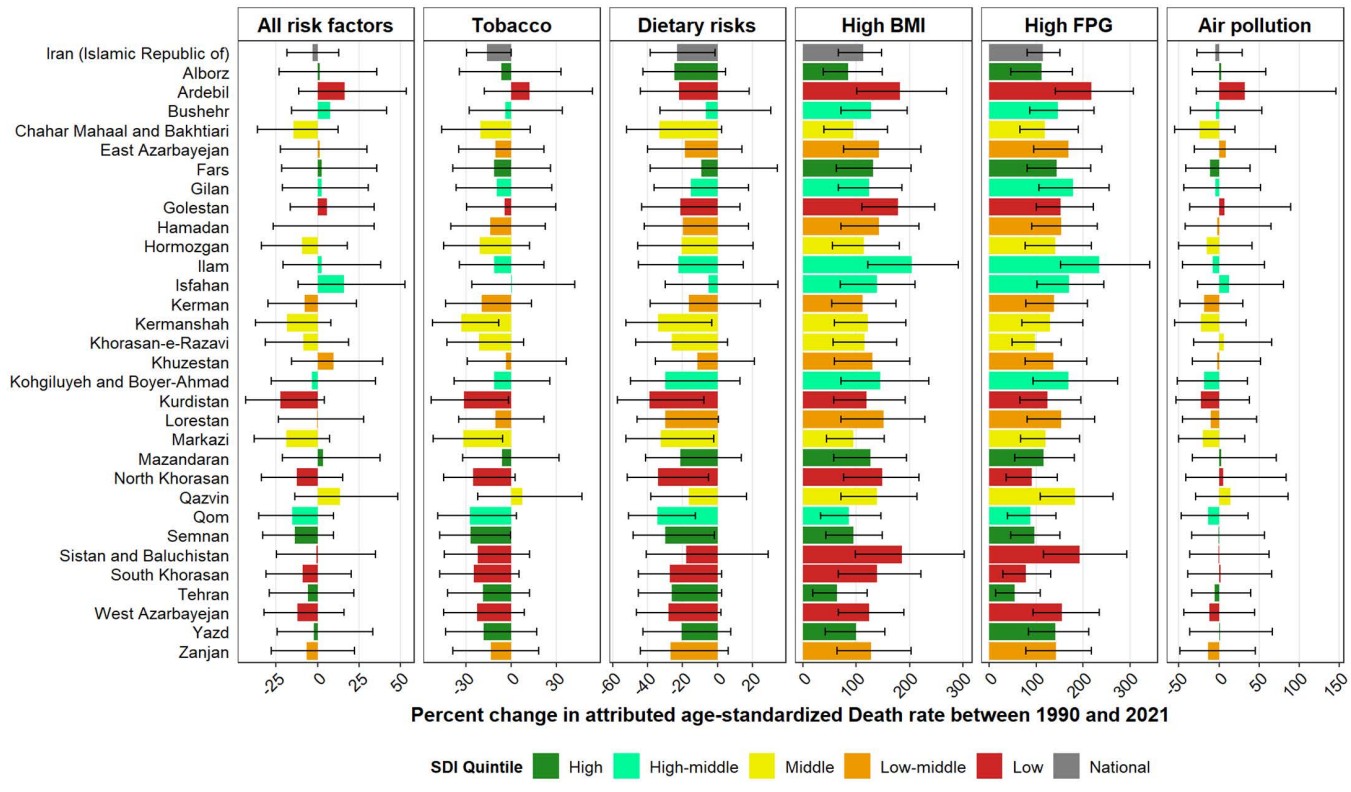

**Fig 4. Percent change of age-standardized death rate of cancer attributable to all and top five level 2 risk factors at the national and subnational levels in Iran from 1990 to 2021, categorized by SDI quintiles.** BMI: Body-mass index; FPG: Fasting plasma glucose.

## 4. Discussion

The present study provided an estimation of the cancer burden attributable to risk factors in Iran from 1990 to 2021, based on the GBD 2021 study. Although age-standardized death and DALY rates attributable to risk factors declined between 1990 and 2021, both rates displayed an upward trend throughout most of the intervening years. The net reduction in risk-attributable age-standardized DALY and death rates may largely be explained by their substantial decline in 2020, potentially linked to the COVID-19 pandemic's impact on delayed cancer screening and diagnosis [25]. During the COVID-19 pandemic, many cancer screening programs were disrupted, primarily because of lockdown measures and the requirement to reallocate healthcare resources. In addition, a reduced quality of data collection and reporting, including possible misclassification of the cause of death, might have resulted in underestimation of cancer burden in Iran throughout the pandemic [26]. Also, a higher proportion of cancer DALYs and deaths was attributable to the estimated risk factors in 2021 compared to 1990. The risk factor-attributable cancer burden was higher in males than in females within all years during the study period. Generally, risk-attributable cancer burden increased with aging, with the highest DALY rate in the 75–79 age group and the highest death rate in the > 80 age group in 2021. Iran shares its leading cancer risk factors, namely tobacco smoking, dietary risks, and obesity, with many other countries in the North Africa and Middle East (NAME) region [27].

While behavioral risks remained the leading cancer risk factor in Iran in all years from 1990 to 2021, the role of metabolic risks in cancer-related burden has become more prominent in recent years. Despite the past, metabolic risks have a greater contribution to cancer burden than environmental/occupational harms. Previous estimations have revealed that

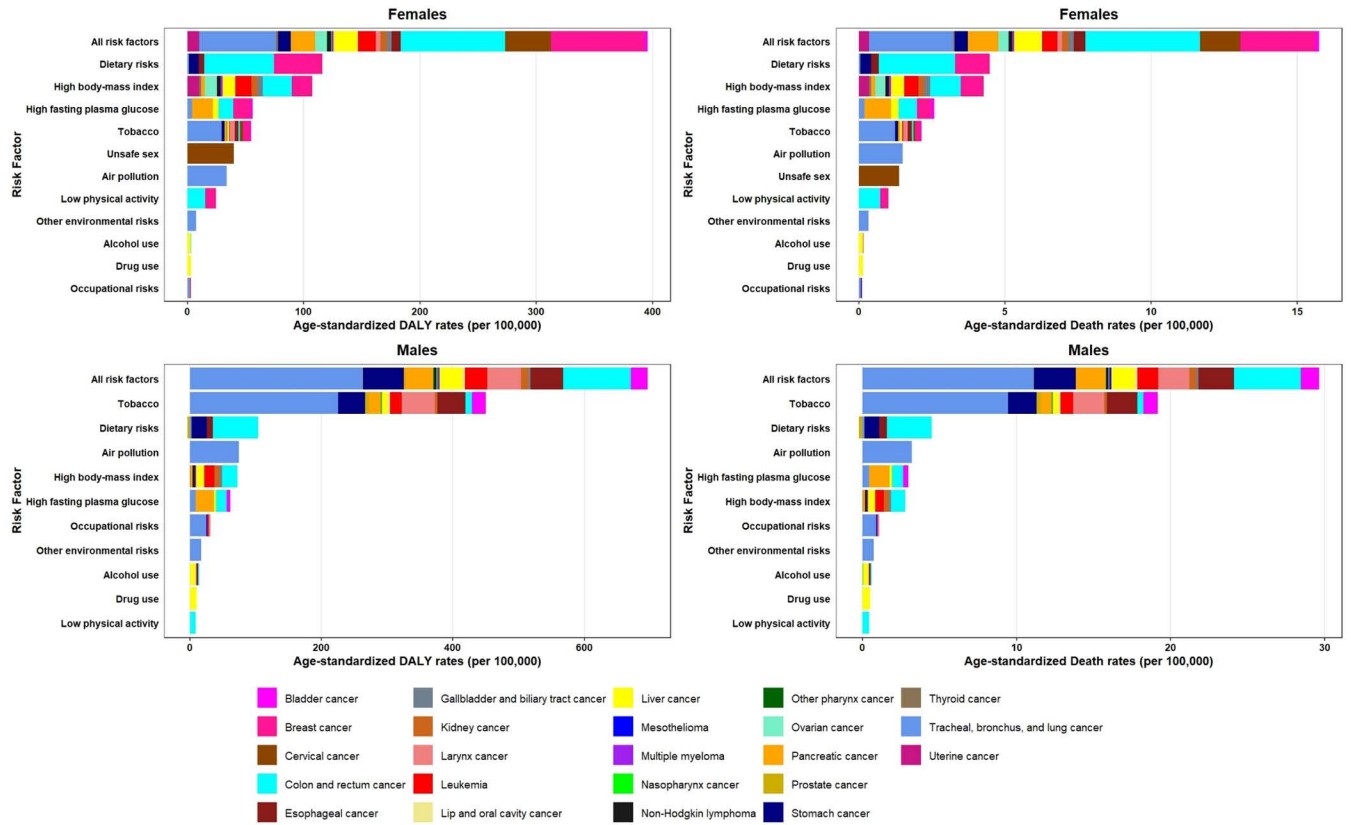

**Fig 5. Age-standardized DALY and death rates of cancer attributable to all and level 2 risk factors by cancer type for females and males in Iran in 2021.**

the cancer burden attributable to metabolic risk factors is surging worldwide [20]. The global age-standardized cancer DALY rates attributable to high FPG and high BMI increased by less than a fifth from 1990 to 2019 [28]; nevertheless, the age-standardized cancer DALY rate in Iran for high FPG and high BMI each increased over twofold, indicating that Iran is facing a greater surge than the global average. In addition, the proportion of cancer DALYs attributable to high BMI and FPG was higher in provinces with higher SDI values. The shift toward urbanization and, hence, lifestyle alteration within the developing nations, including Iran, has contributed to an increase in non-communicable disease burden, such as cancers [29].

Based on our findings, high BMI was the leading metabolic risk for cancer burden in females and males; however, an analysis of the Tehran Lipid and Glucose study revealed that high BMI alone does not increase cancer incidence in Iranian females and males. Instead, a notable elevation in cancer risk occurs when high BMI is accompanied by other metabolic risks, such as high FPG and high systolic blood pressure (SBP) [30]. Therefore, the rising impact of high BMI and high FPG as significant risk factors on cancer burden in Iran demands urgent attention and policy intervention. Iran enacted the Sugar-Sweetened Beverages (SSBs) taxation program in 2013, which was revised in 2021, imposing a 16% and 36% tax on domestic and imported SSBs, respectively; however, household expenditure on SSBs has not declined in Iran despite taxation [31]. This is while other countries in the Eastern Mediterranean region, such as Saudi Arabia, have successfully reduced SSB consumption through taxation [32]. Iran's lower taxation rates, lack of consistent enforcement, and inefficient allocation of tax revenues have limited the effectiveness of its SSB taxation program compared to Saudi Arabia [33]. In

order to overcome these, implementing a strongly enforced higher taxation rate and redirecting tax revenues toward sub-sidizing fruits and vegetables are strongly recommended policy options in Iran's context [31].

Since 1990, there has been a reduction in age-standardized rate of cancer YLLs attributed to behavioral risks, while YLDs have increased. This observation might be derived by advances in cancer survival in Iran over time due to enhanced therapies and increased healthcare access [18]. Also, the survival rates of some leading cancers associated with behavioral risks, including breast cancer, have improved in Iran, although it is still lower than developed countries [34]. From 1990 to 2021, the percent change in the cancer burden of drug and alcohol use was markedly greater in males than females, with males exhibiting a two-fold increase in attributable DALY and death rates. This might largely be explained by the higher prevalence of opium use in males than in females in Iran. A study conducted in Tehran, the capital

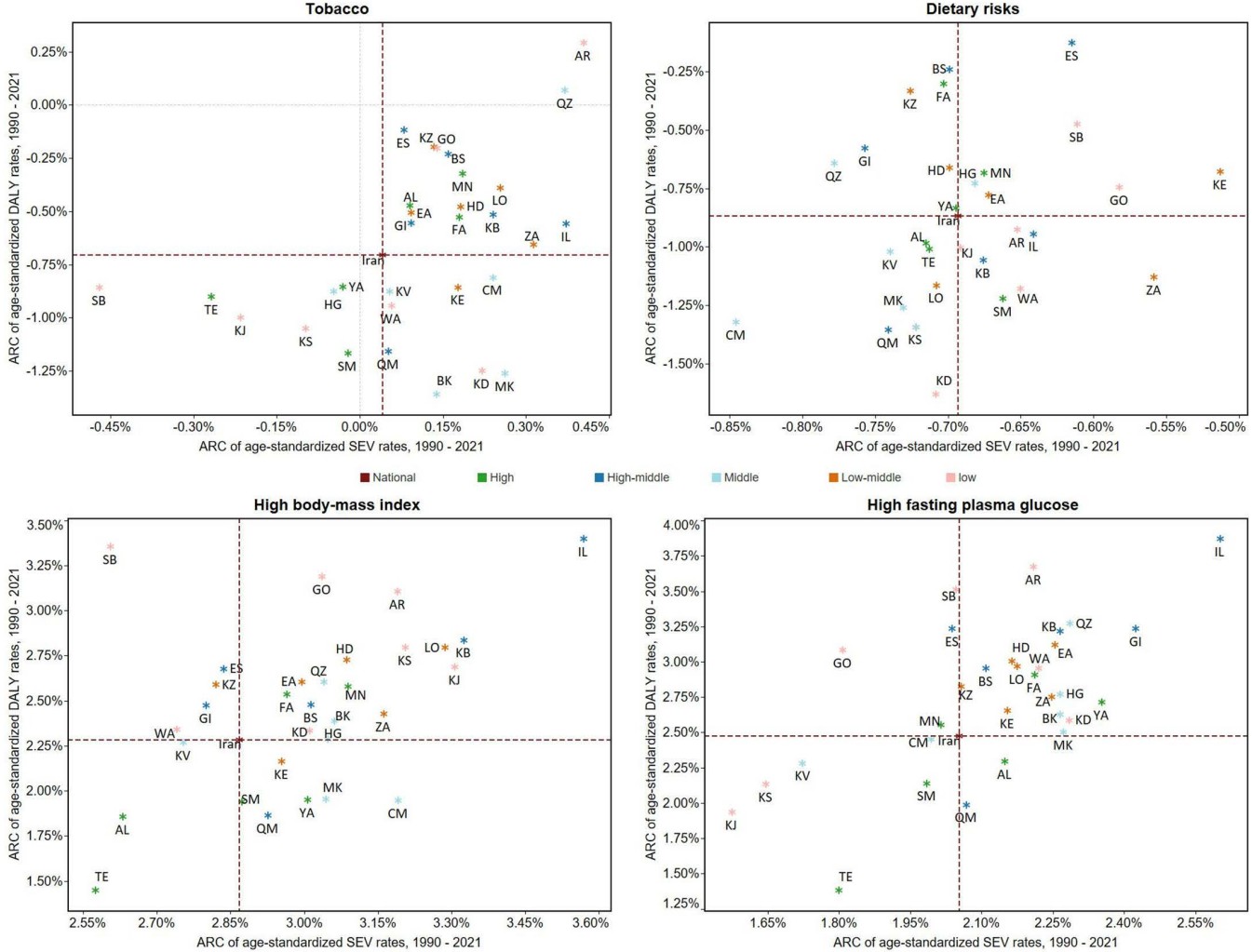

**Fig 6. The ARC of age-standardized DALY rates of cancer attributable to top 4 level 2 risk factors by ARC of their age-standardized SEV rates at the national and subnational levels in Iran from 1990 to 2021, categorized by SDI quintiles.** AL: Alborz; AR: Ardebil; BS: Bushehr; CM: Chahar Mahaal and Bakhtiari; EA: East Azarbayejan; FA: Fars; GI: Gilan; GO: Golestan; HD: Hamadan; HG: Hormozgan: IL: Ilam; ES: Isfahan; KE: Kerman; BK: Kermanshah; KV: Khorasan-e-Razavi; KZ: Khuzestan; KB: Kohgiluyeh and Boyer-Ahmad; KD: Kurdistan; LO: Lorestan; MK: Markazi; MN: Mazandaran; KS: North Khorasan; QZ: Qazvin; QM: Qom; SM: Semnan; SB: Sistan and Blauchistan; KJ: South Khorasan; TE: Tehran; WA: West Azarbayejan; YA: Yazd; Zanjan: ZA.

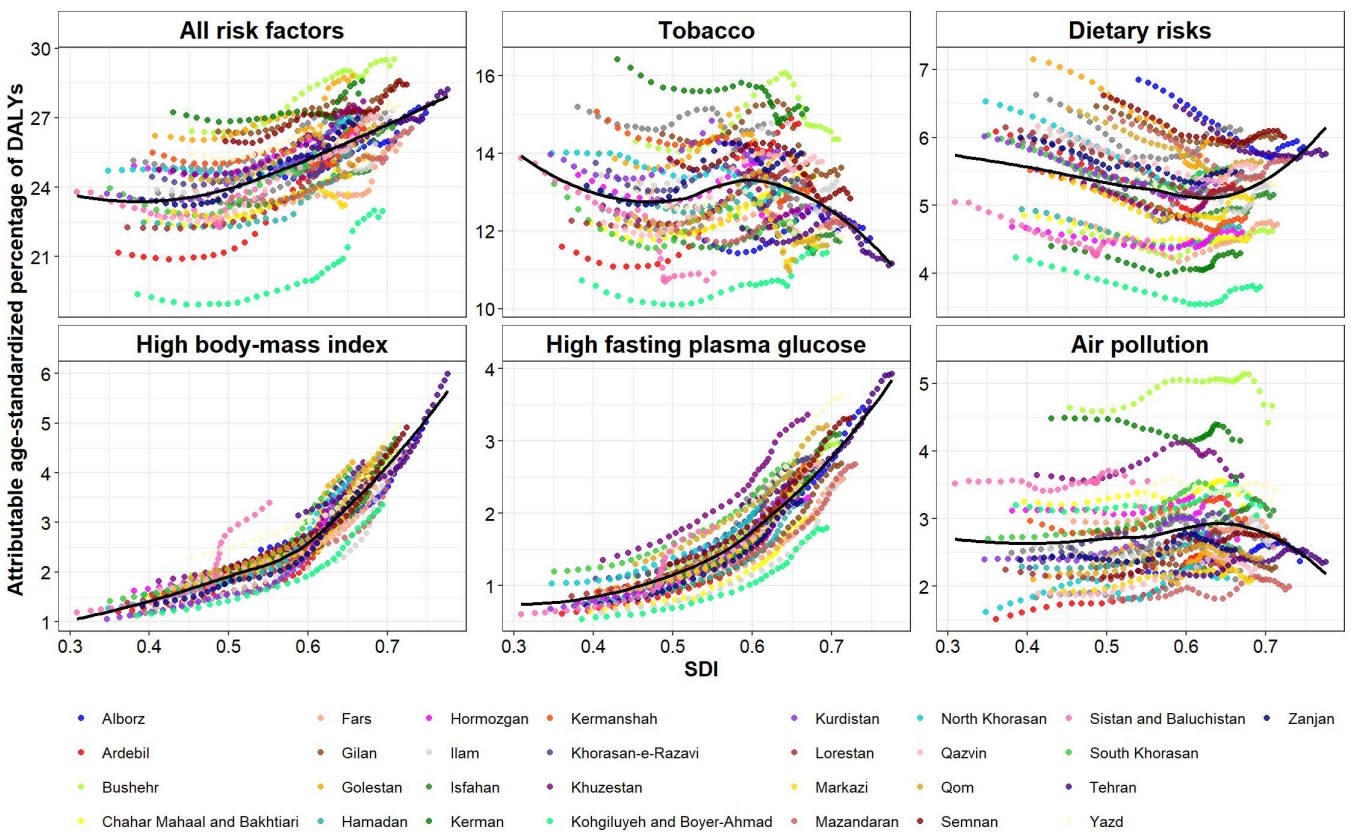

**Fig 7. The relationship between SDI and attributable age-standardized percentage of cancer DALYs for all and the top-five risk factors at the subnational level.**

city of Iran, indicated that the prevalence of opium use was 20 times greater in males than females; additionally, opium and alcohol are consumed in significant association with each other [35]. Opium has been announced as carcinogenic to humans by the International Agency on Research for Cancer (IARC) since 2020 [36]. Previous investigations have attempted to delineate further the association between opium use and various cancers among the Iranian population; accordingly, opium use was found to be associated with gastric [37–39], bladder [39,40], pancreatic [41,42], lung [39,43], colorectal [44], and head and neck cancers [45]. The surge in opium use among Iranian males can be attributed to a combination of factors, including longstanding cultural tolerance of opium and socioeconomic deprivation, such as unemployment and low education [46–48]. Also, Iran's geopolitical position along the drug trafficking routes and domestic economic instability further contribute to widespread availability and use [49].

According to Iran's STEPwise approach to risk factors Surveillance (STEPS) 2021 study, 4.44% (4.09–4.82) and 25.88% (25.03–26.75) of females and males were tobacco smokers, respectively [50]. In the present study, the highest risk-attributable cancer death rate in 2021 was observed in West Azarbayejan, the province with the highest amount of cigarette smoking in the STEPS 2021. Our findings emphasize that although tobacco was the leading cancer risk factor at the national and provincial level in 2021, the tobacco-attributable cancer burden declined in Iran from 1990 to 2021. This decline can be attributed to the fact that Iran has adopted the WHO's proposed measures against tobacco, including monitoring tobacco use, protecting from tobacco smoke, offering help to quit, warning about the harms, enforcing bans, and raising taxes, known as the MPOWER, and was one of the leading Eastern Mediterranean countries in terms of tobacco

smoking reduction in the 2010–2023 years [51,52]. Based on the MPOWER progress report in 2023, Iran is just one measure behind achieving all measures at the best-practice level and is one of the eight countries to reach this milestone out of the 182 countries that have adopted the WHO Framework Convention on Tobacco Control (FCTC); accordingly, Iran has wholly achieved all MPOWER goals except for mass media campaigns against tobacco [53]. Therefore, Iran will benefit from emulating prosperous countries' media policies to reduce tobacco smoking prevalence. Brazil, an outstanding model case in combating tobacco smoking, is the leading country in tobacco use prevalence reduction [54]; Brazil's anti-tobacco media policies include banning tobacco advertisement except at enclosed points of sale and prohibiting tobacco brands' sponsorship of public events [55].

As shown by our results, West Azarbayejan and Ardebil, two provinces in the northwest of Iran, had the highest risk-attributable cancer death rates in 2021. Likewise, a previous spatial analysis of lung, gastric, and esophageal cancer mortality related to smoking and dietary risk factors in Iran identified the northwest of the country as a high-risk area. It noted West Azarbayejan and Ardebil as the most affected provinces by risk-related cancer mortality during 2013–2015 [56]. Additionally, a secondary exploration of the Iran STEPS 2016 data spotted the northwestern part of Iran, encompassing West Azarbayejan, Ardebil, and Gilan, as the area with the highest tobacco- and diet-associated incidence of colorectal cancer [57], the second leading cancer in terms of risk factor-attributable burden in Iran in 2021. Preceded by Ardebil, Golestan, located in the northeast of Iran, was the second leading province regarding risk-attributable cancer DALYs. Results from the Golestan Cohort Study have provided exhaustive information on the major risk factors of cancer in this province throughout more than 14 years, highlighting the impact of tobacco and opium use [39,58,59] and dietary risks [59–61]. Importantly, the cancer burden is projected to continuously grow in Golestan province [62], calling for policy interventions, including risk factor prevention and screening programs, to mitigate cancer incidence in this high-risk area.

From 1990 to 2021, the age-standardized DALY rate of ovarian cancer attributable to all risk factors increased the most compared to other cancers. Accordingly, reports from the Iran National Cancer Registry suggest an increasing trend in 2009–2014 [63]. Of note, high BMI and occupational asbestos exposure were the only risk factors related to ovarian cancer in the GBD 2021 study [20]. High BMI is a major risk for ovarian cancer worldwide [64]; in 2021, 68% of the Iranian female adult population had a BMI of ≥ 25 [65]. Thyroid cancer was the second top cancer regarding the increase in risk-attributable DALY rate between 1990 and 2021, followed by multiple myeloma. In the GBD 2021 study, the sole risk factor associated with thyroid cancer and multiple myeloma was high BMI. According to the previously published GBD 2019 data, the burden of thyroid cancer attributable to high BMI has increased in all provinces since 1990 in Iran [66]. Hence, the top three cancers with the highest increase in risk-attributable DALY rate shared high BMI as their primary risk factor, which further raises concern about the increasing trend of high BMI-attributable cancer burden in the country. Effective policy interventions to reduce BMI among Iranian adults should focus on integrating primary healthcare services for population-level weight management and reducing inequalities in healthy lifestyle behaviors [67].

This study is the first to assess the temporal trends in cancer burden attributable to risk factors over a three-decade interval, utilizing the most updated GBD data. Nevertheless, this study inherits limitations of the GBD methodology. GBD estimates rely heavily on statistical modeling using national input data when available. However, high-quality and updated data, mainly from the Iran National Cancer Registry, may not be frequently incorporated into the GBD estimation methods. While subnational estimates were provided in this study, they might be biased in provinces with sparse data, affecting the accuracy and relevance of the subnational estimations. Another limitation is that the GBD 2021 utilizes universal relative risks for cancer risk factors, which may reduce the reliability of the attributable burden estimates specific to the Iranian population. In addition, the PAFs and cancer burden were estimated in the same year, disregarding the time required for a risk factor to contribute to cancer incidence [28]. The sub-classifications of risk factors and cancer types in the GBD 2021 data are not comprehensive, limiting the ability to include pathologically distinct cancer subtypes and to adequately represent etiologically significant risk factors, such as viral infections. Of note, this study captured trends until 2021, not accounting for changes in risk factor-attributable cancer burden during the post-pandemic era.

## 5. Conclusions

Risk-attributable cancer burden in Iran had an overall upward trend in the last 32 years. Males exhibited a higher risk-attributable cancer burden in Iran throughout all of the study period. The attributable burden was relatively higher in the elderly age groups. While the cancer burden attributable to major behavioral risk factors, including tobacco and dietary risks, has declined, the burden of metabolic risk factors such as high FPG and high BMI has increased tremendously. Among the behavioral risk factors, drug and alcohol use are emerging threats of cancer in the male population. Geographically, the northern and northwestern provinces in the country harbored the greatest burden of risk-attributable cancer; of note, targeted interventions in these regions should align with their cultural and infrastructural aspects. These results shed light on the potential targets of future policies and action plans aiming to mitigate the cancer burden in the country. Our findings provide a comprehensive understanding of the local trends and relevant risk factors, an essential step prior to policy making and strategy implementation. As cancer prevention has proven the most effective way to curtail the rising cancer burden, future strategies focused on the alleviation of cancer risk factors should be given high priority by health policymakers.

## Supporting information

**S1 Table. DALYs, deaths, YLDs, and YLLs of cancer attributable to level 2 risk factors by sex in Iran in 1990 and 2021 and their percent change**. DALYs: Disability-Adjusted Life Years. YLDs: Years Lived with Disability. YLLs: Years of Life Lost.
(DOCX)

**S2 Table. DALYs, deaths, YLDs, and YLLs of neoplasms and level 3 cancers attributable risk factors among females, males, and both sexes in Iran for the years 1990, 2021, and their percent change.** GBD 2021 did not estimate any burden attributable to risk factors for the following 11 level 3 cancers: Brain and central nervous system cancer, Eye cancer, Hodgkin lymphoma, Malignant neoplasm of bone and articular cartilage, Malignant skin melanoma, Neuroblastoma and other peripheral nervous cell tumors, Non-melanoma skin cancer, Other malignant neoplasms, Other neoplasms, Soft tissue and other extraosseous sarcomas, and Testicular cancer. The risk-factor-attributable burden of cervical cancer, ovarian cancer, and uterine cancer was not estimated for males, and the risk-factor-attributable burden of prostate cancer was not estimated for females. DALYs: Disability-Adjusted Life Years. YLDs: Years Lived with Disability. YLLs: Years of Life Lost.
(DOCX)

**S3 Table. DALYs, deaths, YLDs, and YLLs of cancer attributable to all and level 1 risk factors by sex at the subnational level in Iran in 1990 and 2021 and their percent change.** DALYs: Disability-Adjusted Life Years. YLDs: Years Lived with Disability. YLLs: Years of Life Lost.
(DOCX)

**S1 Fig. Trends of age-standardized DALY rates of cancer attributable to all and level 2 risk factors by gender in Iran from 1990 to 2021.**
(DOCX)

**S2 Fig. Trends of age-standardized percentages of cancer DALYs, deaths, YLDs, and YLLs attributed to risk factors in Iran from 1990 to 2021.**
(DOCX)

**S3 Fig. Trends of the percentage of age-standardized cancer DALY rates attributable to each level 2 risk factor relative to cancer DALY rate attributable to all risk factors in Iran from 1990 to 2021.**
(DOCX)

**S4 Fig. The percentage of age-standardized cancer DALY rates attributable to each level 1 risk factor relative to cancer DALY rate attributable to all risk factors by gender at the national and subnational levels in Iran in 1990 and 2021.**
(DOCX)

**S5 Fig. The percentage of age-standardized cancer DALY, death, YLD, and YLL rates attributable to each level 2 risk factor relative to those attributable to all risk factors at the national and subnational levels in Iran in 2021.**
(DOCX)

## Author contributions

**Conceptualization:** Shaghayegh Khanmohammadi, Yasaman Etemadi, Sina Azadnajafabad, Bagher Larijani.

**Formal analysis:** Seyede Maryam Mousavi, Sobhan Younesian, Saba Katebian.

**Investigation:** Saba Katebian, Ali Golestani, Shaghayegh Khanmohammadi, Sepehr Khosravi, Nazila Rezaei.

**Methodology:** Seyede Maryam Mousavi, Sobhan Younesian, Ali Golestani, Sepehr Khosravi, Yasaman Etemadi, Sina Azadnajafabad, Bagher Larijani.

**Project administration:** Ali Golestani, Nazila Rezaei, Sina Azadnajafabad.

**Supervision:** Sina Azadnajafabad, Bagher Larijani.

**Visualization:** Seyede Maryam Mousavi, Sobhan Younesian.

**Writing – original draft:** Seyede Maryam Mousavi, Sobhan Younesian.

**Writing – review & editing:** Saba Katebian, Ali Golestani, Shaghayegh Khanmohammadi, Sepehr Khosravi, Yasaman Etemadi, Nazila Rezaei, Sina Azadnajafabad, Bagher Larijani.

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
