## [Decision Letter · Decision Letter 0]

19 May 2025

Dear Dr. Azadnajafabad,

Thank you for submitting your manuscript to PLOS ONE. After careful consideration, we feel that it has merit but does not fully meet PLOS ONE’s publication criteria as it currently stands. Therefore, we invite you to submit a revised version of the manuscript that addresses the points raised during the review process.

**ACADEMIC EDITOR:**

Thank you for submitting your manuscript to PLOS ONE. The peer-review process has been completed. The reviewers have requested some major issues that need to be addressed before the manuscript can be considered for publication. The detailed feedback from reviewers is included below/attached for your reference.

We kindly request that you address these points in your revised manuscript and provide a response letter detailing the changes made. Please submit the revised version of your manuscript along with the response letter through our submission system.

If you have any questions or require clarification regarding the reviewers' comments, please feel free to contact us. We look forward to receiving your revised manuscript.

We look forward to receiving your revised manuscript.

Kind regards,

Claudio Alberto Dávila-Cervantes, Ph.D.

Academic Editor

PLOS ONE

Reviewers' comments:

Reviewer's Responses to Questions

**Comments to the Author**

1. Is the manuscript technically sound, and do the data support the conclusions?

Reviewer #1: Yes

Reviewer #2: Yes

2. Has the statistical analysis been performed appropriately and rigorously?

Reviewer #1: Yes

Reviewer #2: N/A

3. Have the authors made all data underlying the findings in their manuscript fully available?

Reviewer #1: Yes

Reviewer #2: Yes

4. Is the manuscript presented in an intelligible fashion and written in standard English?

Reviewer #1: Yes

Reviewer #2: Yes

Reviewer #1: The authors reanalyse data collected by the Global Burden of Disease 2021 study to provide an overview of the impact of cancers attributable to risk factors on life expectancy and quality of life in Iran. They show a sharp increase since 1990 in all preventable cancers, across all risk types.

The analysis and model used are appropriate for what the authors set out to do. However I have reservations regarding the correctness of the model that I would like to see addressed:

Major:

- Figure 5 shows quantification of DALY and death rates for ovarian cancer in males. On top of that this quantification is negative for dietary risks. As ovarian cancers do not happen in males (and would not be significantly detectable even including trans-men) this is likely an artifact of the model used which should be corrected.

- While the data used for the analysis is publicly available, the code used to perform it is not which prevents the evaluation of the correctness of the analysis. Please make the code available with the manuscript.

Minor:

- typos: COIVD

- The text of some cells in Table 1 is truncated (e.g. all DALY cells). Please extend the cells to fix it.

- I would like to see a discussion of the decrease in YLL but increase in YLD of behavioral risks seen in Table 1.

Reviewer #2: The study utilizes GBD to analyze the trends of cancer burden attributed to risk factors in Iran between 1990 and 2021.

While the study has merits, I think it would benefit from addressing these points.

1. It is not clear how the study is different than reframing the GBD data. The authors need to highlight how their study adds novel insights rather than just GBD.

I am not sure what are the original analyses that were done beyond the GBD

2. The methods section is basically re-writing the GBD methods. No information about what “authors” have done; version? Which R scripts? Date accession? This will ensure reproducibility.

3. How local MIRs were derived

4. The introduction reads like separate paragraphs that are not linked to each other. It needs to be tied around one idea and needs to flow seamlessly.

5. There are differences between ecological modeling and causation. The authors have to be very clear when saying that this risk factor cause X DALYs

6. Have the authors applied FDR in their statistical analysis for multiple comparisons?

7. In figure 3 the x axis min and max are large making it hard to see the differences and the intervals

8. Figure 4 have abbreviations that I assume they are countries names. But no reader will be familiar of all these abbreviations

9. The colors in Figure 5 are very similar to each other for example, bladder cancer and uterine cancer are both having the same shade of blue.

10. There are no mention of study limitations and GBD limitations at all in the discussion. This is very crucial.

11. How was percent change was calculated because there are no UIs in table 1?

12. Analyzing the subnational level data is very important so it would benefit from exploring the reasons behind these differences. A map with matrix plot like colors for showing the differences in risk factors and DALYs would be beneficial to the manuscript (spatial map)

13. Why the authors attribute the reduction to COVID19?

14. The discussion would benefit from comparing Iran to the other similar countries to should the similarities and the differences

15. Exploring the reasons behind this surge in opium use is important in the discussion

16. Table one seems to have been cut in the first couple of rows (the upper UI is not found.) for example all age numbers 169057.47 (135238.74 to )

17. Risk-attributable cancer burden in Iran had an overall upward trend” contradicts earlier claim that ASDR declined

18. The discussion section is overwhelming while it does not have a deep, evidence-based discussion linking findings to actionable policies.

**Do you want your identity to be public for this peer review?** For information about this choice, including consent withdrawal, please see our Privacy Policy

Reviewer #1: No

Reviewer #2: No

---

## [Author Response · Author response to Decision Letter 1]

6 Jun 2025

Dear Editorial Team and Esteemed Reviewers,

We sincerely thank the editors and reviewers for their valuable comments and insightful suggestions, which have significantly improved our manuscript both methodologically and in terms of presenting our findings. Your constructive feedback allowed us to refine critical aspects of the study, including the refinement of the Introduction, the clarification and strengthening of the Methods, improvements in the figures, and enhancement of the Discussion. We have provided a detailed response to each comment in the following sections of this letter. Additionally, we have revised the manuscript using tracked changes for your convenience. We hope these updates meet your expectations and facilitate the review process. Thank you once again for your thoughtful and meticulous review.

Reviewer #1 comments:

Comment: Figure 5 shows quantification of DALY and death rates for ovarian cancer in males. On top of that this quantification is negative for dietary risks. As ovarian cancers do not happen in males (and would not be significantly detectable even including trans-men) this is likely an artifact of the model used which should be corrected.

Response: We sincerely thank the reviewer for carefully pointing out this issue. The problem with Figure 5 was a mislabeling in the legend. We have recreated the figure to correct this error. The estimated DALY and death rates for prostate cancer attributable to dietary risks in males are negative in the dataset.

Comment: While the data used for the analysis is publicly available, the code used to perform it is not which prevents the evaluation of the correctness of the analysis. Please make the code available with the manuscript.

Response: Thanks for your comment! The code used to perform all estimations used in this manuscript are publicly available at https://github.com/ihmeuw/ihme-modeling/tree/main/gbd_2021. We have also added the link to the manuscript, emphasizing that the code is also publicly available. We have also clearly stated the version of R (version 4.4.0) and Python (version 3.12.4) used and specified the date on which the GBD data were accessed (October 20, 2024) in the manuscript. Additionally, we would like to clarify that Tableau, which was used for some data visualizations, does not require coding scripts. Other visualizations were generated using standard procedures from the ggplot2 package in R and pandas library in Python, and LOESS regression was implemented using base R function [loess (val ~ sdi)].

Comment: Typos: COIVD

Response: Thanks! We have corrected this fault in the text.

Comment: The text of some cells in Table 1 is truncated (e.g. all DALY cells). Please extend the cells to fix it.

Response: Thanks for noticing that. We have extended the cells vertically and horizontally so that the whole UI interval can now be seen.

Comment: I would like to see a discussion of the decrease in YLL but increase in YLD of behavioral risks seen in Table 1.

Response: Thanks for your suggestion. In response we have a added a discussion of this observation to the Discussion section in the revised manuscript, which reads: “Since 1990, there has been a reduction in age-standardized rate of cancer YLLs attributed to behavioral risks, while YLDs have increased. This observation might be derived by advances in cancer survival in Iran over time due to enhanced therapies and increased healthcare access. Also, the survival rates of some leading cancers associated with behavioral risks, including breast cancer, have improved in Iran, although it is still lower than developed countries.”

Reviewer #2 comments:

Comment 1: It is not clear how the study is different than reframing the GBD data. The authors need to highlight how their study adds novel insights rather than just GBD.

Response: We appreciate your thoughtful feedback. We respectfully contend that our study offers substantial value beyond just re-reporting GBD data and contributes to understanding the cancer burden attributable to risk factors in Iran at both national and subnational levels. Here, we outline the key aspects that distinguish our work and underscore its significance. First, Iran currently lacks sufficient peer-reviewed publications comprehensively reporting the burden of cancers attributable to risk factors, particularly at the subnational level. Our study addresses this gap by providing a detailed analysis of cancer types and attributable risk factors across Iran’s provinces from 1990 to 2021. This will inform localized public health policies, which are often constrained by the absence of such data in the Iranian context.

Moreover, our study enhances the utility of GBD data by developing original visualizations that elucidate temporal patterns and spatial variations in risk-attributable cancer burden over 32 years. These visualizations, tailored to highlight trends and patterns, offer an accessible and exploratory tool for policymakers, extending beyond the raw data outputs typically provided by GBD. Including such figures represents a novel contribution, enabling a deeper interpretation of the data not readily available in the GBD database.

Also, you’re right that our initial analysis did not include heavy statistical methods. We have since incorporated Locally Estimated Scatterplot Smoothing (LOESS) regression analysis to examine the relationship between SDI and the percentage of attributable cancer DALYs for top five risk factors at the subnational level. We have also employed the annualized rate of change (ARC) analysis to enhance the interpretation of data. The ARC analysis assessed the relationship between the annual change in summary exposure values (SEV) for specific risk factors and the corresponding annual change in cancer disability-adjusted life years (DALYs) attributable to those risk factors at the subnational level, with estimates derived for the period 1990 to 2021. The findings from this analysis are illustrated in Figure 6, which highlights the top four risk factors and categorizes subnational regions by SDI quintiles, providing deeper insights into the underlying drivers of these trends. Additionally, the LOESS regression modeled the non-linear association between SDI and the percentage of attributable DALYs for specific risk factors, with results presented in Figure 7. This analysis, incorporating data from all provinces over the 1990–2021 period, reveals that the proportion of cancer DALYs attributed to high body-mass index and high fasting plasma glucose increases with rising SDI values. These novel findings elucidate the relationship between SDI and the contribution of risk factors to cancer burden. Therefore, our study includes several analyses.

Finally, our study's originality lies in its contextual relevance and synthesis. By focusing on Iran, a country with a unique demographic, epidemiological, and socio-economic profile, we critically adapt global data to a national and subnational framework. This localized perspective is vital for translating global estimates into actionable insights, a step that GBD alone cannot achieve without region-specific interpretation and analysis.

We believe that these contributions, including filling a literature gap, enhancing data through visualizations, and applying statistical methods, demonstrate that our study is far more than a re-reporting of GBD data. Instead, it represents a valuable extension that supports evidence-based decision-making in Iran’s public health landscape. We hope this clarification addresses your concerns and reinforces the merit of our work.

Comment 2: The methods section is basically re-writing the GBD methods. No information about what “authors” have done; version? Which R scripts? Date accession? This will ensure reproducibility.

Response: Thanks for pointing out this issue. You are correct that the initial version of the Methods section devoted considerable space to describing the GBD 2021 methodology, with relatively limited emphasis on our own analytical procedures. In response, we have revised the section to provide a more concise summary of the GBD methodology and have placed greater focus on the analyses we conducted, including the calculation of the annualized rate of change (ARC) and the application of LOESS regression. We have also clearly stated the version of R (version 4.4.0) used and specified the date on which the GBD data were accessed (October 20, 2024) in the manuscript. Additionally, we would like to clarify that Tableau, which was used for some data visualizations, does not require coding scripts. Other visualizations were generated using standard procedures from the ggplot2 package, and LOESS regression was implemented using base R function [loess (val ~ sdi)]. The results of LOESS analysis are depicted in the Figure 7, which has been newly added to our manuscript in the revised version

Comment 3: How local MIRs were derived

Response: Thanks for your question. We’ll provide relatively detailed answer here based on the GBD 2021 methodology documentation. However, we have also added a few sentences clarifying your question in the manuscript. Accordingly, cancer incidence and mortality data from registries were matched by cancer type, age, sex, year, and location to calculate mortality-to-incidence ratios (MIRs), which were used as inputs for further modeling. For most cancers, MIRs were estimated using a three-step spatiotemporal Gaussian process regression (ST-GPR) approach, where logit-transformed MIRs were modeled with covariates including sex, categorical age groups, and the Healthcare Access and Quality (HAQ) Index in a linear mixed-effects model, followed by spatiotemporal smoothing and Gaussian process regression. The ST-GPR model employed fixed hyperparameters (lambda = 0.05 for time, omega = 0.5 for age, zeta = 0.01 for geography, amplitude = 1, and scale = 10) to smooth data across dimensions, with manual outlier removal for unrealistic data points. For rare cancers in young age groups, data were aggregated to the youngest five-year age bin with at least 50 cases from 1990–2015 SEER data, with MIRs applied to younger age groups. MIRs were capped at the 95th percentile by age group to allow values above 1, with pediatric age groups (under 20) capped at 1 for flexibility; values exceeding caps were Winsorised, and inputs were scaled for ST-GPR and rescaled post-modeling, with lower caps at the fifth percentile to constrain underestimates. This methodology ensures robust and flexible MIR estimates across diverse cancers and demographics.

We have briefly explained this in the Methods section: “MIRs were estimated using cancer registry data matched by cancer type, age, sex, year, and location. A spatiotemporal Gaussian process regression (ST-GPR) model incorporated covariates such as age, sex, and the Healthcare Access and Quality Index, with smoothing across time, age, and geography. Additionally, adjustments were made for rare cancers and outliers to ensure reliable estimates across all cancer types and demographic groups.”

Comment 4: The introduction reads like separate paragraphs that are not linked to each other. It needs to be tied around one idea and needs to flow seamlessly.

Response: Many thanks for your valuable insight! We have revised and reframed the introduction section to improve the coherence between paragraphs. Changes can be tracked in the “Revised Manuscript with Track Changes” file.

Comment 5: There are differences between ecological modeling and causation. The authors have to be very clear when saying that this risk factor cause X DALYs

Response: Thanks for pointing out this important consideration! We have explicitly avoided the use of the word “caused by” while referring to the cancer burden attributed to risk factors and have used the word “attributed to” instead. There was one instance in the manuscript that we wrote “cancer DALYs due to metabolic risks”, which we have now edited and replaced it with “cancer DALYs attributed to metabolic risks”. It is also worth mentioning that in order to be consistent with the GBD terminology, we have used the term “causes of death” to refer to the cancer types, as this is the standard GBD terminology.

Comment 6: Have the authors applied FDR in their statistical analysis for multiple comparisons?

Response: Thanks for your insightful question! Multiple comparisons are a major issue when the model involves many statistical tests and thus multiple comparisons, as this is the case in our work. In order to provide an answer, we should delve deeper into the GBD 2021 methodology. GBD 2021 relies on Bayesian hierarchical models, meta-regression (e.g., MR-BR), and ensemble modeling (e.g., DisMod-MR) rather than FDR for handling multiple comparisons. These methods focus on uncertainty intervals (UIs) from Bayesian posterior distributions or bootstrap methods to quantify variability. Therefore, FDR is not applied, but this aligns with the study’s reliance on Bayesian methods, which inherently handle multiple comparisons through hierarchical modeling and uncertainty quantification rather than p-value adjustments. For instance, in estimating risk factor-attributable burden, mediated-adjusted population attributable fractions (PAFs) were used to account for correlations between risk factors, suggesting a different approach to managing multiple testing issues than FDR.

Comment 7: In figure 3 the x axis min and max are large making it hard to see the differences and the intervals

Response: Thanks for your helpful feedback. We had used fixed scale for the Figure 3, which made it difficult for some values to be seen. We have now recreated the Figure 3 and used free scales in the x-axis for each risk factor. Now the percent changes and intervals can be more easily observed. Given that the X-axis scale is now tailored to each risk factor rather than being uniform across all graphs, it is important to interpret comparisons between risk factors with caution, as the differing scales may affect visual interpretation. Of note, because we now have added a spatial map plot, the name of the figure we are discussing in this comment has changed to Figure 4 in the revised manuscript.

Comment 8: Figure 4 have abbreviations that I assume they are countries names. But no reader will be familiar of all these abbreviations

Response: You are absolutely right in highlighting this issue. These abbreviations stand for the names of Iran’s provinces. As these names are too long, it was not possible to include the full names of the provinces in the figure. We have added the explanation for these abbreviations in the caption of the Figure 3.

Comment 9: The colors in Figure 5 are very similar to each other for example, bladder cancer and uterine cancer are both having the same shade of blue.

Response: Thanks for your valuable feedback. We have recreated the Figure 5, utilizing more distinct colors to improve clarity.

Comment 10: There are no mention of study limitations and GBD limitations at all in the discussion. This is very crucial.

Response: We truly appreciate your attention to this crucial point. We have added a comprehensive paragraph discussing the study’s limitations and GBD limitations at the end of the Discussion section in the manuscript, which reads:

“This study is the first to assess the temporal trends in cancer burden attributable to risk factors over a three-decade interval, utilizing the most updated GBD data. Nevertheless, this study inherits limitations of the GBD methodology. GBD estimates rely heavily on statistical modeling using national input data when available. However, high-quality and updated data, mostly from Iran National Cancer Registry, may not be frequently incorporated into the GBD estimation methods. While subnational estimates were provided in this study, they might be biased in provinces with sparse data, affecting the accuracy and relevance of the subnational estimations. Another limitation is that the GBD 2021 utilizes universal relative risks for cancer risk factors, which may reduce the reliability of the attributable burden estimates specific to the Iranian population. In addition, the PAFs and cancer burden were estimated in the same year, disregarding the time required for a risk factor to contribute to canc

---

## [Decision Letter · Decision Letter 1]

7 Aug 2025

Temporal trend in the national and sub-national burden of cancers attributable to risk factors in Iran from 1990 to 2021: findings from the Global Burden of Disease Study 2021

PONE-D-25-12699R1

Dear Dr. Azadnajafabad,

We’re pleased to inform you that your manuscript has been judged scientifically suitable for publication and will be formally accepted for publication once it meets all outstanding technical requirements.

Kind regards,

Claudio Alberto Dávila-Cervantes, Ph.D.

Academic Editor

PLOS ONE

Additional Editor Comments (optional):

Reviewers' comments:

Reviewer's Responses to Questions

**Comments to the Author**

Reviewer #1: All comments have been addressed

2. Is the manuscript technically sound, and do the data support the conclusions?

Reviewer #1: Yes

3. Has the statistical analysis been performed appropriately and rigorously?

Reviewer #1: Yes

4. Have the authors made all data underlying the findings in their manuscript fully available?

Reviewer #1: Yes

5. Is the manuscript presented in an intelligible fashion and written in standard English?

Reviewer #1: Yes

Reviewer #1: (No Response)

**Do you want your identity to be public for this peer review?** For information about this choice, including consent withdrawal, please see our Privacy Policy

Reviewer #1: No

---

## [Editor Report · Acceptance letter]

PONE-D-25-12699R1

PLOS ONE

Dear Dr. Azadnajafabad,

I'm pleased to inform you that your manuscript has been deemed suitable for publication in PLOS ONE. Congratulations! Your manuscript is now being handed over to our production team.

Kind regards,

on behalf of

Mr. Claudio Alberto Dávila-Cervantes

Academic Editor

PLOS ONE